# Association between Dietary Factors and Constipation in Adults Living in Luxembourg and Taking Part in the ORISCAV-LUX 2 Survey

**DOI:** 10.3390/nu14010122

**Published:** 2021-12-28

**Authors:** Maurane Rollet, Torsten Bohn, Farhad Vahid

**Affiliations:** Nutrition and Health Research Group, Department of Population Health, Luxembourg Institute of Health, l-1445 Strassen, Luxembourg; maurane.rollet@orange.fr (M.R.); farhadvahid@outlook.com (F.V.)

**Keywords:** macronutrients, micronutrients, diarrhea, digestive diseases, food groups, colon

## Abstract

Constipation, a disorder of bowel movements, is among the most frequent gastrointestinal complaints in Western countries. Dietary constituents such as inadequate fiber intake have been related to constipation, but discrepancies exist in the findings regarding dietary factors. This study investigated the association between dietary patterns and bowel movements in adults living in Luxembourg. Data from 1431 participants from ORISCAV-LUX 2 (a cross-sectional survey) who completed a 174-item food frequency questionnaire (FFQ) were analyzed. A questionnaire-based constipation score was assessed by a validated scoring system. Confounders such as physical activity and serum/urine indicators were assessed. Women had higher constipation scores than men (*p* < 0.001). In food group-based regression models, a negative association was found between higher constipation score and intake of grains (Beta = −0.62, 95%CI: −1.18, −0.05) and lipid-rich foods (Beta = −0.84, 95%CI: −1.55, −0.13), while a positive association was found for sugary products (Beta = 0.54, 95%CI: 0.11, 0.97) (*p* < 0.05). In a nutrient-based regression model, a positive association was found between constipation score and total energy (Beta = 5.24, 95%CI: 0.37, 10.11) as well as sodium intake (Beta = 2.04, 95%CI: 0.21, 3.87), and a negative one was found for total fats (Beta = −4.17, 95%CI: −7.46, −0.89) and starch (Beta = −2.91, 95%CI: −4.47, −1.36) (*p* < 0.05). Interestingly, neither fruits and vegetables or dietary fiber were significantly associated with constipation. Thus, grains, lipid-rich foods, total fats and starch were associated with a lower constipation score, while sugary products, sodium, and higher energy intake were correlated with higher constipation.

## 1. Introduction

Bowel movements describe the frequency and comfort of stool production, i.e., the fecal discharge. They are an essential process by which the body eliminates undigested and unabsorbed food constituents along with cells and microflora [1]. Feces is characterized by its frequency, type, and contents [2]. A frequency of bowel movements between three per week and three per day is typically defined as “normal” defecation [3].

A large-scale multinational study based on internet surveys has shown that >40% of people worldwide suffer from gastrointestinal complications such as diarrhea, constipation, or irritable bowel syndrome (IBS), with a prevalence of 4.7%, 11.7%, and 4.1%, respectively [4]. Bowel movements have also been related to gastrointestinal diseases such as inflammatory bowel disease (IBD), Crohn’s disease, and ulcerative colitis [5]. As such, extreme bowel movements (constipation or high frequency) could be early signs of gastrointestinal disorders or other diseases [6].

Various factors such as gender, age, and body mass index (BMI) have been related to bowel movements [7]. Women have been reported to be more frequently impacted by bowel movement-related disorders, especially chronic constipation [8]. However, increasing age is related to increased gastrointestinal motility disorders for both genders, i.e., constipation, diarrhea, or incontinence [9]. Disorders related to body weight including anorexia [10] and morbid obesity [11] are likewise related to bowel movement complaints. Apart from these disorders, many other factors have been related to irregular bowel movements, including perceived stress, anxiety, and depression [12], changes in the gut microbiota [13], infections [14], physical inactivity [15], and altered sleeping patterns [16].

Along with other factors, dietary factors such as dietary patterns/habits, food group consumption preferences, and macro- and micronutrient intake have been most typically reported to modulate the activity of the gastrointestinal tract [17]. For instance, the amount of calories in a meal has been related to the digestive tract colonic motor response [18], which also impacts the stomach [19] and the small intestine [20]. Low energy intake may be associated with constipation by slowing down the colonic transit time [21]. Excess calorie intake may likewise deregulate the proper functioning of the gastrointestinal tract [22]. Low liquid, i.e., water intake from food and beverages, has also been associated with constipation, potentially because of the osmotic action of fluids [23]. Due to their impact on fecal bulk [24] and in part via acting as prebiotics [24], higher dietary fiber intake has been related to more frequent bowel movements.

Because nutrients are able affect gastrointestinal food passage, it is unsurprising that dietary patterns have been related to bowel movements [24]. The Westernized diet is characterized by a large amount of fat, principally saturated and trans fats, along with high intake of refined sugar, excess salt consumption, and limited intake of fruits, vegetables, and dietary fiber [25]. Contrarily, it has been reported that the Mediterranean diet significantly improves bowel movements due to its plant-rich profile [26]. In line with this report, people following vegetarian and/or vegan diets have higher bowel movement frequency than meat and/or fish-eaters [7]. In general, high consumption of whole grains, fruits, vegetables, nuts, and seeds promotes bowel movements [27]. Conversely, high consumption of fast food, junk food, and/or processed food has been associated with functional gastrointestinal disorders [28]. A few specific dietary patterns are also employed as dietary therapies to treat functional bowel disorders, including lactose-free, gluten-free, low-carb, or low-FODMAP (fermentable oligosaccharides, disaccharides, monosaccharides, and polyols) diets [29].

Limited studies have been conducted investigating the effects of macronutrients other than carbohydrates on bowel movements, and even less so for included micronutrients. In the present investigation, we aimed to study food groups, macro- and micronutrients, and non-nutrients (i.e., energy and fiber) and their relation to bowel movements in adults residing in Luxembourg and attending the ORISCAV-LUX 2 survey, as a first step toward estimating the relationship between these factors and gut health.

## 2. Materials and Methods

### 2.1. Study Population and Design

The analyses were based on the study ORISCAV-LUX 2—Observation of Cardiovascular Risk Factors in Luxembourg 2. Details about the study design have been previously published [30]. Briefly, ORISCAV-LUX 2 is the second cross-sectional, nationwide survey on the prevalence of cardiovascular risk factors in the Luxembourgish adult population [31]. Carried out in 2016–2017, it is a follow-up to the ORISCAV-LUX study, conducted in 2007–2008 in Luxembourg [32]. A total of 1558 Luxembourgish residents aged 25–81 years were enrolled in this study. However, the study protocol stated that people could enter the study until the age of 79. As this study was a continuation of the first wave taking place ten years earlier, a number of participants in the second wave were over 79 years old, and were therefore excluded from analyses. Therefore, 1431 individuals were retained for this analysis, i.e., they delivered a complete dataset including the nutritional aspects (Figure 1).

### 2.2. Data Collection

Data collection included data from questionnaires related to sociodemographic aspects, lifestyle, and self-reported health conditions including a validated food frequency questionnaire (FFQ) [33]. We calculated a constipation score based on data related to bowel movements. Two questionnaires were employed: a home-based self-reported questionnaire with 117 items and a second questionnaire administered by the study nurse with 135 items. The home-based questionnaire was principally a self-assessment of the individual’s lifestyle and environment. In this questionnaire, the participants had to self-report their bowel movements and well-being by answering eight questions related to bowel movements. The nurse questionnaire was more focused on the practical evaluation of mental and physical health, including the FFQ. Anthropometric and clinical measurements were also assessed, and the nurse scheduled appointments at a private laboratory for blood and urine sample collection and analysis. In the present study, we focused on selected variables previously cited in the scientific literature as associated with bowel movements (Table 1, Table 2 and Table 3). All individuals were appropriately informed and consented to participate in the survey. The study was approved by the National Research Ethics Committee (CNER) and the National Commission for Data Protection (CNPD).

### 2.3. Measurement of Constipation Score

A previously published constipation scoring system (Table 1) was applied as a valid tool for evaluating constipation in the study participants [34]. This system was based on a study involving patients with constipation, and the questionnaire was validated against clinical physiological measurements such as colonic transit time. This score was calculated based on the eight questions from this home-based questionnaire, which included questions about the frequency of bowel movements, the frequency of painful evacuation, the use of any type of assistance (such as laxatives or enema), time spent on the lavatory per attempt, the number of unsuccessful attempts per day, the frequency of incomplete evacuation, the recurrence of abdominal pain, and the duration of constipation (in years). Depending on the answers, the scores ranged from zero to 30, with zero indicating a normal bowel movement, 15 indicating constipation, and 30 equaling severe constipation.

### 2.4. Assessment of Dietary Habits

The participants completed an online questionnaire under the supervision of the nurse, a validated quantitative FFQ [32] used to collect data about dietary habits [35]. The quantity and frequency of consumption of 174 food and beverage items were recorded to determine dietary intake. A score ranging from ‘never or rarely’, ‘one to three times/month’, ‘one to two times/week’, ‘three to five times/week’ ‘once a day’, to ‘twice or more a day’ was used to estimate food frequency consumption, and portion size images were used to report food quantity. From these outputs, we merged the items into twelve food groups based on their nutritional characteristics: grains, starchy vegetables, fruits, vegetables, protein rich-food, ready-made meals, dairy, lipids, sugary products, non-caloric beverages, sugared sweetened beverages, and alcoholic beverages.

In addition, the macronutrient and micronutrient intakes were obtained by converting food and beverages into nutrients using the French Ciqual food database, which lists the nutritional composition of >3100 food items [36]. The total energy was calculated by summing up fat (9 kcal/g), alcohol (7 kcal/g), protein (4 kcal/g), carbohydrates except for polyols (4 kcal/g), organic acids (3 kcal/g), polyols (2.4 kcal/g), and dietary fiber (2 kcal/g). In this way, we provided a general overview of the dietary patterns of adults living in Luxembourg, who were participants in the ORISCAV-LUX 2 survey, including food groups and macro- and micronutrients.

### 2.5. Assessment of Physical Activity

Physical activity as a potential confounder was assessed using accelerometers (ActiGraph^TM^ GT3X+ 3D-accelerometer, Pensacola, FL, USA). Instructions were given during the appointment with the nurse. Participants had to wear the bracelet on the wrist day and night for seven consecutive days. Physical activity was assessed during this period by measuring acceleration motion in three different axes of the body (the vertical, mediolateral, and anteroposterior axes) [37]. We principally considered the results from the accelerometer on intensity gradient and average acceleration, which corresponded to the distribution of activity intensities across 24 h and the average intensity over the day, respectively [38].

### 2.6. Measurement of Blood/Urine Markers

Blood samples were drawn after overnight fasting, and urine samples were collected as early morning midstream urine specimens [39]. Samples were stored in the Integrated BioBank of Luxembourg, and analyses were later performed by a commercial accredited company (Ketterthill, Esch-sur-Alzette, Luxembourg). From the collected blood samples, we obtained blood concentrations of C-reactive protein (CRP), thyroid-stimulating hormone (TSH), free triiodothyronine (FT3) and free thyroxine (FT4) hormones, testosterone, estradiol, insulin, and glycated hemoglobin (HbA1C) as well as serum levels of sodium, magnesium, potassium, and vitamin D. From the spot urine samples we used only urinary sodium concentration.

### 2.7. Data Management

Of all the enrolled participants (ORISCAV-LUX 2 = 1558), only those who had completed the FFQ were considered for further analysis. In this respect, 127 (8.1%) participants from ORISCAV-LUX 2 were excluded due to missing FFQ data. The variables for which data were missing were accelerometer data (20.8%), blood samples (up to 4.3%), and urine samples (8.5%). In this case, the missing answers were noted as “did not answer”. Regarding sociodemographic variables with missing data (highest for income), we grouped them into “did not answer” as well in order to analyze the data. Therefore, no participants were excluded from the analysis due to missing data in sociodemographic variables. In total, 1431 participants completed the FFQ and were included in our analyses.

### 2.8. Statistical Analyses

#### 2.8.1. General Aspects

Statistical analyses were performed using SPSS (IBM, Chicago, IL, USA) v. 25. The normality of data and equality of variance was tested by Q–Q plots/Kolmogorov–Smirnov test and boxplots. If data were not normally distributed, log-transformation was attempted. Unless otherwise reported, all data are median ± interquartile ranges (IQR). T-tests were employed for continuous variables and the Chi-square test was used for categorical variables (all continuous variables were log-transformed). The primary endpoint for analysis was the constipation score. We investigated the relation of constipation score to dietary intakes based either on nutrients or on food groups. *p*-values below 0.05 (2-sided) were considered statistically significant. In addition, the Benjamini–Hochberg correction was applied to all *p*-values; all *p*-values are displayed following this correction.

#### 2.8.2. Bivariate and Partial Correlations

Bivariate correlation analyses were carried out based on Pearson correlations, employing either log-transformed (originally non-normally distributed) or original (normally distributed) data. In the following, partial correlation analyses were carried out controlling for major confounding factors. Bivariate correlation analyses were also carried out, stratified for sex, age, and BMI.

#### 2.8.3. Multivariable Linear and Logistic Regression Modeling

Regression modeling in SPSS included a forward/stepwise selection as well as a backward elimination procedure, which was finally chosen from among the different methods for regression analysis. For linear regression models, *p*-values below 0.1 (two-sided) were considered for inclusion in the next step or as means for elimination. This process enabled obtaining a model from the full set of variables (saturated model) by automatically removing those that did not contribute significantly to the model (step-down procedure). In order to ensure that no significant variables were missed in our model, we chose to undertake an additional check by manually inserting the most significant variables one-by-one (step-up procedure). The regression strategy is further described in Figure 2.

Due to the limited number of food groups, all were initially included in the analyses of the food groups. Variables retained in the final model were grains, vegetables, lipids, and sugary products, and retained confounders were education level, job, testosterone serum level, and sodium serum level.

For nutrients, we selected parameters based on bivariate correlations that were significantly correlated with constipation score or parameters reported in the literature related to bowel movements for nutrients and confounders. We selected total energy intake, vegetable protein intake, total fat intake, polyunsaturated fat intake, starch intake, fiber intake, alcohol intake, vitamin C, magnesium, and potassium intake. Confounders in both models included age, gender, BMI, income, education, and marital status. In addition, concentrations of CRP, free T3, free T4, estradiol level, testosterone, vitamin D, and serum sodium were selected as blood-based markers. Variables retained in the final model included total energy intake, total fat intake, starch intake, sodium intake, and as confounders BMI, education level, job, testosterone serum level, and sodium serum level (Figure 2).

In addition, we performed logistic regression analyses based on constipation score categories (quartiles) for all food groups and macro-and micronutrients. All models were adjusted for education level, job, and serum levels of sodium and testosterone.

Analyses were also stratified by sex, age, BMI, job, and education, i.e., analyses were done separately for men and women. In addition, age was divided into five subgroups (<34.99/35–44.99/45–54.99/55–64.99/65 years and more) based on the WHO age classification [40]. BMI (in kg/m^2^) was categorized as underweight–normal (<24.99), overweight (25–29.99) and obese (>30) [41]. Job status was divided into four categories, employed, unemployed, leave/retired, and did not answer. Education level was sorted into highest educational degree obtained (primary level, secondary level, post-secondary level, and did not answer).

## 3. Results

### 3.1. Description of the Study Participants and Dietary Patterns

Table 2 shows the anthropometric, socioeconomic, and blood marker characteristics of participants based on gender and constipation score categories (quartiles). There was a significant difference between men and women in terms of BMI, education, job, income, constipation scores, average daily time spent in vigorous physical activity, and testosterone and sodium serum concentrations. In addition, there was a significant difference between constipation score quartiles regarding income and testosterone serum level (Table 2).

In addition to serum levels of testosterone and sodium, other biological measures, e.g., serum level of CRP, free T3, free T4, estradiol, and vitamin D (Figure 2), were also investigated; however, these were not found to have any significant correlation and were not included in final analyses. The median age of the participants was 51 years, of whom 53.2% were women. The frequency charts of constipation score by age group for both men and women are shown in Figure 3.

The medians and interquartile range values of daily intake of food groups (Table 3) and nutrients (Table 4) are presented based on gender and constipation score categories. Men consumed significantly higher amount of grains, starchy vegetables, protein rich foods, ready-made meals, lipids (fats and oils), sugary products, and alcoholic beverages than women, while the latter consumed more vegetables than men (Table 3). In addition, participants in quartiles (Q) 1, 2, 3 (combined) consumed significantly higher amount of grains and alcoholic beverages compared to participants in Q4 (Table 3). Conversely, participants in the Q4 significantly consume a higher amount of fruits than participants in Q1, 2 and 3 (combined) (Table 3). Moreover, participants in Q1, 2 and 3 (combined) consumed significantly higher amounts of starch and alcohol compare to participants in Q4 (Table 4) while the latter consumed significantly higher amounts of vitamin C (Table 4). Men also consumed generally higher amounts of nutrients than women.

### 3.2. Bivariate and Partial Correlation Analyses

A significant negative correlation between constipation score and individual nutrients/non-nutrients was found for total energy intake, vegetable protein, total fats, PUFA, starchy vegetables, and lipids (all *p*-values < 0.05, Table 5). A significant negative correlation was found for intake of starch, alcohol, grains, and alcoholic beverages (all *p*-values < 0.01).

Significant positive correlations with constipation score were found for intake of vitamin C among nutrients, and for fruits and vegetables among food groups (all *p*-values < 0.05).

Additional significant correlations were found for subgroups divided by gender, BMI, and age group (Table 5), i.e., fruits only in men and vegetables only in women, grains in age group 4 (55–64.99 years), and BMI group 1 (<24.99 kg/m^2^), among others.

### 3.3. Multivariable Linear and Logistic Regression Modeling

The final food group-based model revealed a significant negative association between constipation score and intake of grains and lipid foods. A significant positive association was found between sugary products, and a positive trend with vegetables. The total R for this model was 0.253 (Table 6).

In the nutrient-based model, a significant positive association between constipation score and total energy and sodium intake and a significant negative association with total fats and starch was found in the final retained model. The total R for this model was 0.258 (Table 7).

In the logistic regression analyses (adjusted for education level, job, serum levels of sodium and testosterone) a significant association was observed between the intake of fruits and the odds of a higher constipation scores (OR = 1.48, 95%CI: 1.02–2.15; *p*-value = 0.008). There was also a non-significant association (trend) between vitamin C intake and the odds of a higher constipation score (OR = 1.66, 95%CI: 0.98–2.83; *p*-value = 0.060), possibly due to the relationship between vitamin C and fruit intake. Regarding other food items and macro- and micronutrients, as they were insignificant in the logistic models, results are not presented.

## 4. Discussion

In the present study, we investigated the relationship between constipation and dietary habits. Based on our findings, a limited number of the investigated dietary aspects were significantly correlated with constipation scores. Based on the final retained multivariable regression model, these included grains, lipid-rich foods, and sugary products from among the food groups, and various nutrients that were significantly associated with constipation score either positively (sugary products, total energy, and sodium intake) or inversely (grains, lipids, total fats, starch intake). According to the multivariable linear regression model, the strongest contributors appeared to be total energy and total fat intake. These findings were also observed in bivariate correlations.

Studies have previously reported that the amount of calories in a meal can directly influence gastrointestinal passage, and this may therefore be associated with constipation [18,20]. Towers et al. showed that lower energy intake could be associated with constipation due to slow colonic transit time [21] related to a lower fecal bulk. On the other hand, Delgado-Aros et al. showed that excessive calorie intake might likewise deregulate proper digestion and cause disturbances such as constipation [22]. Similar to our results, their study showed a higher constipation prevalence in the population with higher caloric intake. It is possible that high-energy intake was related to energy-dense food items such as processed foods and ready-made meals, which have been previously related to constipation [42].

Although dietary lipids have been reported to increase colonic myoelectric and motor activity in several cases, resulting in more regular bowel movements [43], more frequently and in general a high-fat diet has been associated with gastrointestinal problems, including constipation [44]. For example, Wibisono et al. have reported that constipation is a common side effect of a lipid-rich ketogenic diet [45], though both diarrhea and constipation were reported for this diet by others [46]. Lipids are typically fully absorbed in the small intestine and add very little to the bulk fecal bulk in the colon [44], and lipid calories often replace carbohydrates associated with dietary fiber.

Regular consumption of saturated fatty acids (SFA) has also been associated with constipation due to slower gastrointestinal transit caused by enteric neurodegeneration [47]. SFA can induce changes in gut microbiota composition, potentially leading to intestinal dysbiosis [47]. Conversely, polyunsaturated fatty acids (PUFA) such as omega-3 or long-chain PUFAs can modulate inflammation and the immune system, fostering a healthy symbiosis in the gut bacteria [48]. In line with this, a significant inverse association in bivariate correlations between PUFA intake and constipation score was noted in our study. On the contrary, high intakes of omega-6 PUFAs appear to be implicated in the pathogenesis of IBD, perhaps due to their associated pro-inflammatory properties [49].

Interestingly, in the present study, lipid-rich food items (comprising fats and oils) were inversely associated with constipation scores (Table 5). However, this group contained 27 food items, some of which may act very differently in terms of their effects on the digestive process. For instance, butter and olive oil were both included in this group; however, they have somewhat different compositions and are typically consumed with different meals. For instance, olive oil is often consumed with salads while butter is typically used for various cooking procedures [50].

Although carbohydrate malabsorption can be related to constipation, diarrhea, or IBS [51], simple sugars generally have a minor impact on bowel movements compared to complex carbohydrates, especially dietary fiber [27], as they are almost completely absorbed in the upper intestine. Dietary fiber, by contrast, consists of non-digestible carbohydrates and is recognized as a primary dietary factor influencing bowel movements [27]. At the colon level, the microbiota transform some fibers—the fermentable fraction—into short-chain fatty acids (SCFAs), which aid in the proper functioning of colonic epithelial cells [27]. Some types of fiber can act as prebiotics, which stimulate the growth of gut bacteria related to health benefits for the host [52] and positively shorten colonic transit time [53].

In the present study, the daily intake of grains was related to a lower constipation score. In line with our results, de Vries et al. previously reported that grain consumption was linked to improved digestive processes. Grains, especially whole grains, are a rich source of dietary fiber; they contain up to 10–12 g/100 g [54], increasing fecal bulk and promoting faster intestinal/colonic transit time [55]. Stool consistency is likewise positively affected, decreasing constipation [54]. Similarly, one meta-analysis reported that dietary fiber intake could increase stool frequency, decreasing constipation. However, it appeared that dietary fiber intake did not have an effective impact on severe constipation, and was more effective for mild to moderate forms [56]. However, negative influences of dietary fiber consumption on symptoms of constipation have also been reported. For instance, Ho et al. reported that patients with idiopathic constipation might improve their symptoms by reducing dietary fiber intake in the diet and foods such as grains [57]. Dietary fiber can activate fermentation by the gut microbiota, enhancing bacterial mass [58] and gas formation, and producing volatile compounds such as SCFAs, which can add to digestive discomfort [58]. As a limitation of this investigation, we did not assess the effect of the different types of grains, such as refined grains versus whole grains. Refined grains have relatively low dietary fiber levels, i.e., around 2–3 g/100 g (vs. 10–12 g/100 g for whole grains) [59], are less likely to aid in functional constipation, and are related to a higher prevalence of constipation [60]. However, it must be considered that some carbohydrates, including those in bread, may be present in the form of resistant starches, which would likewise reach the colon and act in a fiber-like fashion [61].

Similar to grain intake, starch intake was associated with a lower constipation score in the present study (both multivariable linear regression and bivariate correlations). Starchy vegetables appeared to have the same impact. Again, the co-existence of dietary fiber (including resistant starch) within these food groups could have played a role. However, our study could not differentiate between resistant and non-resistant starches, as these are typically not listed in food databases. However, it has been hypothesized that countries with a high starch consumption, such as certain African countries and possibly within the Mediterranean diet characterized by high pasta intake, associated acid fecal pH has been explained by the presence of resistant starches [62].

In contrast to starch, sugary products were positively associated with constipation scores in this investigation. The sugary food group was principally composed of items rich in refined sugar, such as jams, ketchup, chocolate, biscuits, snacks, ice cream, gelatin desserts, and candies. In particular, refined sugary products have been implicated in many gastrointestinal disorders [63], especially in constipation, as these are typically low in fiber and often also high in unhealthy fats [36]. Some studies have shown that frequent consumption of sugary snacks was linked to increased odds of constipation [64]. On some occasions, excess intake of certain sugars such as fructose [65] and polyols [65] may be related to very loose stools due to the osmotic effect of these sugars in the gut [65]. However, a relatively high intake of these sugars/sugar alcohols is needed to cause these effects, and the majority of simple sugars in typical Westernized diets are sucrose and glucose [66].

Fruits are also naturally rich in simple sugars; however, it is generally accepted that any adverse effects on the intestine and colon are outweighed by their fiber content, especially pectin, which adds to the bulk of fecal material and facilitates gastrointestinal passage [67]. This is in line with our findings, which showed no negative effects of fruits on constipation, although no positive effects based on the final multivariable regression models were revealed either.

Similar to fruits, and rather surprisingly, vegetable intake did not significantly correlate with constipation in either the bivariate analyses or the multivariable linear regression model. Vegetable intake has been usually linked with decreased constipation, explained mainly by their being rich in dietary fiber [68]. We did not investigate the cooking method employed for any food or nutrients, and it is known that cooking methods may directly influence food digestion [69]. It is also possible that vegetables within the present study were typically consumed as a side dish together with meat or other lipid-rich and calorie-rich food items, thus impeding the finding of significant correlations.

Sodium was the only micronutrient that was connected with altered, i.e., higher, constipation score (multivariable linear regression model, Table 7). Large amounts of salt can decrease the amount of water in the stool due to its osmotic activity being taken up [70], making it more difficult to move along the digestive tract and causing difficulties in bowel movements. In addition, food items rich in sodium comprise especially processed foods, ready-made meals, and meat products [71], which are low in fiber and water. Furthermore, some studies have shown that a high-salt diet might alter the gut microbiota composition and balance, leading to an imbalance between healthy and harmful bacteria [72], which is known to cause gut disorders as a most common consequence [73].

Surprisingly, food constituents usually related to constipation in the scientific literature including dietary fiber [56], water [23], and magnesium [74] were not directly associated with the constipation score in our analysis (multivariable linear regression model and bivariate correlations). Dietary magnesium, for instance, has been related to intestinal disorders such as constipation [74], and magnesium-rich water appeared to be efficient in reducing constipation [75]. We can only speculate on the reasons for this observation. First, although we obtained a large sample of participants, most participants in this survey were rather healthy, i.e., with no particular gastrointestinal complications. Few people reported digestive complaints and high constipation; only about 10 (0.8%) people scored above 15 on constipation score. In addition, in our study, there was no additional focus on gastrointestinal complications besides constipation. Only a few questions were asked to calculate a constipation score, even though this score is recognized in the scientific literature and its association/correlation with biological outcomes was investigated [34]. In addition, one of the limitations of this score is that it does not weigh the questions, and the questions are considered equal with respect to their contribution to the final score [34]. However, any other such weighting could have been regarded as subjective as well, and the originally developed scoring correlated well with observed symptoms, e.g., 98% of constipations were correctly predicted. Perhaps future studies with other designs (validity/reliability/calibration) could answer the question of whether weighting can improve the accuracy of such scores or further elucidate the best basis for weighting questions, which was beyond the scope of the present study. In addition, we did not differentiate between different kinds of constipation, such as slow colonic transit type or obstructed defecation type [76]. Indeed, clinical manifestations of constipation were not assessed, nor were any other digestive issues, e.g., bloating, diarrhea, or irritable bowel syndrome. Finally, although physical activity is often cited in the scientific literature as positively impacting constipation symptoms, our analysis did not reveal any association between physical activity and constipation score in the study population.

## 5. Conclusions

In conclusion, the food groups of grains and lipids as well as total fats and starch were associated with lower constipation scores. Conversely, sugary products, sodium intake and higher energy intake were correlated with higher constipation scores. Surprisingly, some items, which are well-documented to impact constipation, such as fruit and vegetable intake, were not associated with constipation score in our analysis. Further studies are required to investigate in more detail the impact of different dietary profiles on bowel movements. In line with this, the relation of diseases to digestive disorders such as diarrhea, irritable bowel syndrome, and bloating, among others, should be investigated in more detail in order to more readily counteract the potential transition to disease manifestation.

## Figures and Tables

**Figure 1 nutrients-14-00122-f001:**
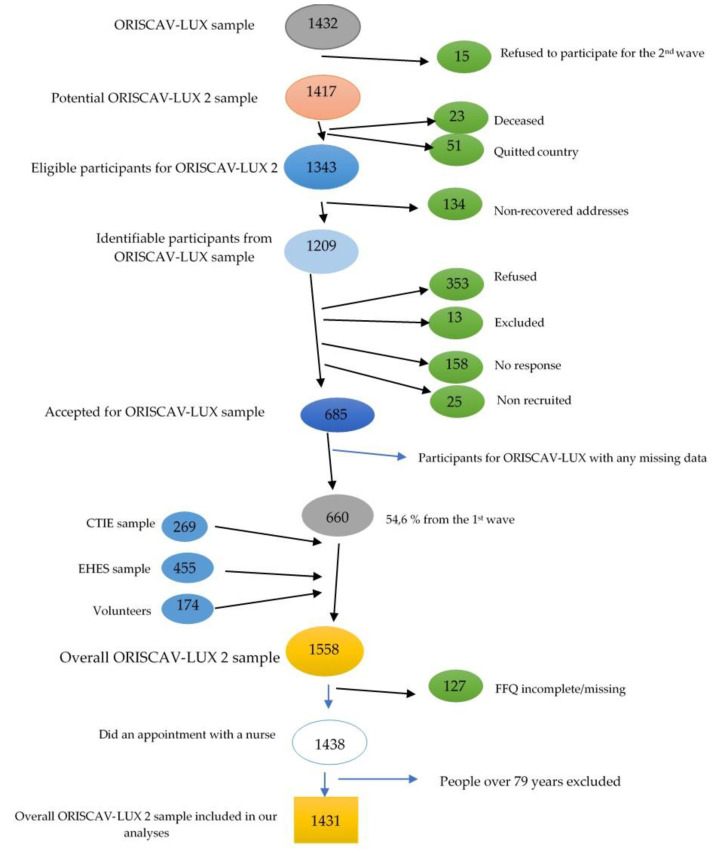
Participant sample progression. Fifteen people refused to participate in the second wave from the outset, lowering the number of potential participants to 1417. When sending the invitations of participation for the second wave, 134 addresses were not found, further decreasing the total number to 1209. Of these identified people, 353 refused to participate, 13 were excluded (moved abroad, physical disability or language incapacity), 25 (non-recruited) did not attend or canceled their repeated appointments, and 158 did not respond to the invitation; thus, 685 people from the first sample accepted participation in the second wave. CITIE = Centre des technologies de l’information de l’Etat, EHES = European Health Examination Survey. FFQ = food frequency questionnaire.

**Figure 2 nutrients-14-00122-f002:**
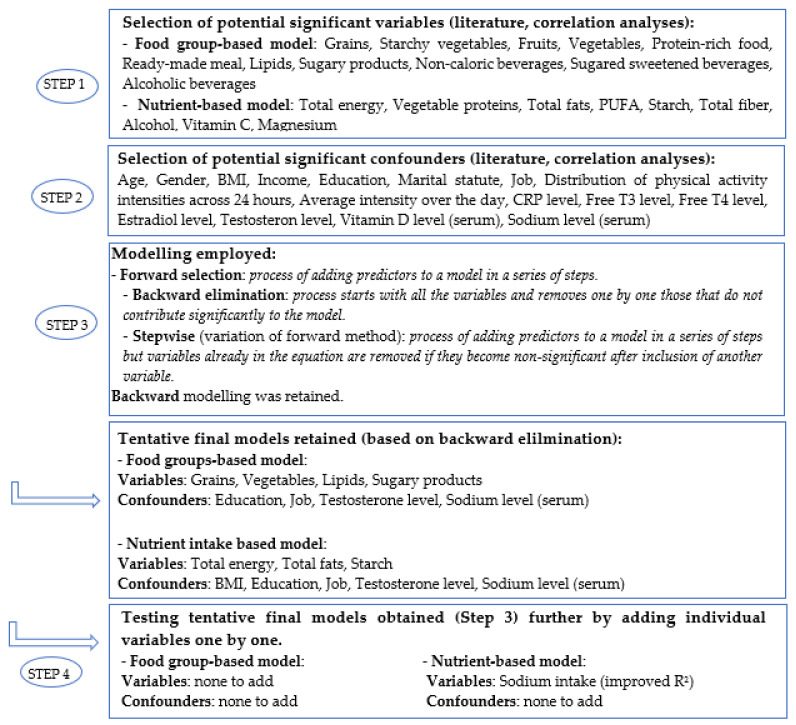
Linear regression progression (backward elimination). For the food group-based model, all food groups were kept in the original model. Nutrients and confounders were pre-selected based on the literature and on bivariate correlation analysis. Variables and confounders were eliminated stepwise in order to keep only significant variables or those where a trend was apparent (*p* < 0.1).

**Figure 3 nutrients-14-00122-f003:**
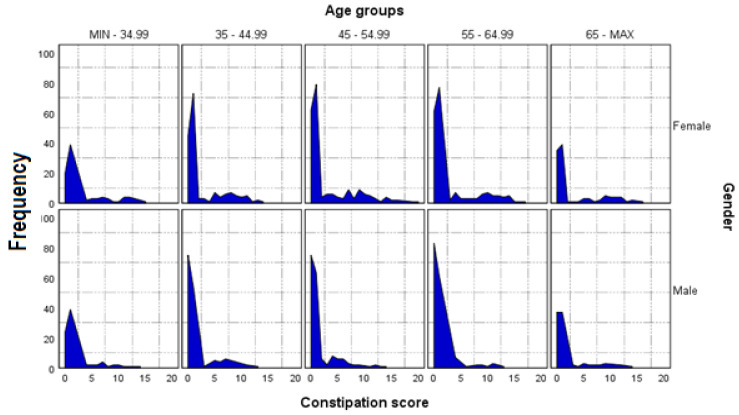
Frequency chart of constipation score by age groups in men and women. Zero is the lowest possible score (no constipation); 30 represents severe constipation. Score according to [34].

**Table 1 nutrients-14-00122-t001:** Questionnaire underlying the constipation-scoring algorithm, modified based on [34].

Question	Answer	Score
Frequency of bowel movements	1–2 times per 1–2 days	0
	2 times per week	1
	once per week	2
	<once per week	3
	<once per month	4
Difficulty of evacuation (painful)	Never	0
	Rarely	1
	Sometimes	2
	Usually	3
	Always	4
Incomplete feeling of evacuation	Never	0
	Rarely	1
	Sometimes	2
	Usually	3
	Always	4
Abdominal pain	Never	0
	Rarely	1
	Sometimes	2
	Usually	3
	Always	4
Minutes spent in lavatory per attempt	<5 min	0
	5–10 min	1
	10–20 min	2
	20–30 min	3
	>30 min	4
Assistance: type of assistance	Without assistance	0
	Simulative laxatives	1
	Digital assistance or aenema	2
Unsuccessful attempts for evacuation/24 h	Never	0
	1–3	1
	3–6	2
	6–9	3
	>9	4
Duration of constipation	0	0
	1–5 years	1
	5–10 years	2
	10–20 years	3
	>20 years	4
Total sum	30	

**Table 2 nutrients-14-00122-t002:** Distribution of demographic, anthropometric, socioeconomic, and blood marker characteristics of participants.

Variables	Median (IQR) or Number (%)						
	Total (*n* = 1431)	Women (*n* = 761)	Men (*n* = 670)	*p*-Value ^ꝉ,^^Ƌ^	CS1 (*n* = 1078)	CS2 (*n* = 353)	*p*-Value ^ꝉ,^^Ƌ^
Age (year)	50.77 (17.99)	51.5 (17.6)	50.1 (18.4)	0.397	50.8 (18.5)	50.8 (16.8)	0.888
BMI (kg/m^2^) - Normal - Overweight - Obesity - Other ^f^	655 (45.8%) 499 (34.9%) 288 (18.7%) 9 (0.6%)	417 (54.8%) 219 (28.7%) 118 (15.5%) 7 (0.9%)	238 (35.5%) 280 (41.8%) 150 (22.3%) 2 (0.3%)	<0.001	495 (45.9%) 371 (34.4%) 206 (19.1%) 6 (0.6%)	160 (45.3%) 128 (36.3%) 62 (17.6%) 3 (0.8%)	0.734
Education - No diploma * - Secondary education ** - Post-secondary education *** - Other ^e^	188 (13.1%) 504 (35.3%) 612 (42.7%) 127 (8.9%)	108 (14.2%) 279 (36.7%) 297 (39.0%) 77 (10.1%)	80 (11.9%) 225 (33.6%) 315 (47.0%) 50 (7.5%)	0.015	130 (12.1%) 378 (35.1%) 474 (44.0%) 96 (8.9%)	58 (16.4%) 126 (35.7%) 138 (39.1%) 31 (8.8%)	0.146
Job - Employed - Unemployed ^a^ - Leave ^c^ - Other ^e^	940 (65.7%) 153 (10.7%) 319 (22.3%) 19 (1.3%)	476 (62.5%) 126 (16.6%) 149 (19.6%) 10 (1.3%)	646 (69.3%) 27 (4.0%) 170 (25.4%) 9 (1.3%)	<0.001	697 (64.7%) 115 (10.7%) 253 (23.5%) 13 (1.2%)	243 (68.8%) 38 (10.8%) 66 (18.7%) 6 (1.7%)	0.271
Monthly income (Euro) - Less than 750 - 750 to 1499 - 1500 to 2249 - 2250 to 2999 - 3000 to 4999 - 5000 to 10,000 - More than 10,000 - Other ^e^	4 (0.3%) 23 (1.6%) 51 (3.6%) 82 (5.7%) 340 (23.8%) 487 (34.0%) 117 (8.2%) 327 (22.9%)	1 (0.1%) 15 (2.0%) 32 (4.2%) 57 (7.5%) 171 (22.5%) 236 (31.0%) 48 (6.3%) 201 (26.4%)	3 (0.4%) 8 (1.2%) 19 (2.8%) 25 (3.7%) 169 (25.2%) 251 (37.5%) 69 (10.3) 126 (18.8%)	<0.001	2 (0.2%) 20 (1.9%) 32 (3.2%) 59 (5.5%) 257 (23.8%) 383 (35.5%) 93 (8.6%) 232 (21.5%)	2 (0.6%) 3 (0.8%) 19 (5.4%) 23 (6.5%) 83 (23.5%) 104 (29.5%) 24 (6.8%) 95 (26.9%)	0.033
Constipation score - 0 to 5 - 6 to 10 - 11 to 15 - >16	1186 (82.8%) 158 (11.0%) 81 (5.7%) 6 (0.5%)	586 (77.0%) 105 (13.8%) 64 (8.4%) 6 (0.8%)	600 (89.6%) 53 (7.9%) 17 (2.5%) 0 (0%)	<0.001			
Physical activity - TSED (minutes/day) - TMPA (minutes/day) - TVPA (minutes/day) - AID (per day)	655.6 (680.5) 54.1 (76.5) 0.7 (2.0) 23.3 (12.4)	637.1 (659.8) 56.3 (79.6) 0.6 (1.7) 24.2 (11.6)	682.3 (696.4) 52.6 (73.7) 0.7 (2.1) 22.2 (12.7)	0.880 0.450 0.022 0.408	656.6 (681.2) 54.4 (77.2) 0.7 (2.0) 23.5 (12.5)	651.4 (670.7) 53.8 (77.3) 0.6 (1.7) 23.1 (12.5)	0.700 0.637 0.238 0.359
Biological measurements Testosterone serum level	0.47 (4.90)	0.27 (0.13)	5.3 (2.6)	<0.001	2.37 (5.18)	0.34 (4.09)	<0.001
Sodium serum level	141.0 (2.0)	141.0 (3.0)	141.0 (2.0)	<0.001	141.0 (3.0)	141.0 (2.0)	0.158

*n* = number of participants included in the analysis, IQR = interquartile range, BMI = body mass index, TSED = Average daily time spent in sedentary behavior, TMPA = Average daily time spent in moderate physical activity, TVPA = Average daily time spent in vigorous physical activity, AID = Average intensity over the day, CS1 = constipation score (Q1 + Q2 + Q3), CS2 = Q4. * Pre-primary and primary education. ** CATP—Certificate of Technical and Professional Aptitude, CITP—Certificate of Technical and Professional Initiation, CCM—Certificate of Manual Capability, Diploma for Completion of Secondary Technical Studies, Diploma for Completion of Secondary General Studies. *** Technician diploma, Bac +2 (BTS), Bac +3 (Bachelors/Degree), Bac +4 (Masters), Bac +5 and more (3rd Cycle, DEA, DESS, MBA, Masters, Ph.D., etc.), Diploma from a Grande Ecole, an Engineering School. ^a^ In school, university, or in training, at home, unemployed, or in search of employment. ^c^ Retired or in early retirement or long-term leave. ^e^ Other, did not know, did not answer. ^f^ missing data. ^ꝉ^ T-test was used for continuous variables and Chi-square test was used for categorical tests. (All continuous variables were log-transformed). **^Ƌ^** Benjamini–Hochberg correction was applied to all *p*-values; all *p*-values are displayed after this correction.

**Table 3 nutrients-14-00122-t003:** Distribution (median and IQR) of food group consumption per day according to gender and constipation score groups.

Variables	Median (IQR)						
	Total (*n* = 1431)	Women (*n* = 761)	Men (*n* = 670)	*p*-Value ^ꝉ,^*	CS1 (*n* = 1078)	CS2 (*n* = 353)	*p*-Value ^ꝉ,^*
Grains (g)	135.9 (119.3)	121.3 (107.5)	155.2 (141.0)	<0.001	137.8 (116.8)	122.9 (123.2)	0.027
Starchy vegetables (g)	57.1 (61.1)	52.8 (62.4)	64.3 (69.4)	<0.001	59.5 (65.0)	53.5 (60.5)	0.069
Fruits (g)	294.6 (281.7)	307.5 (281.0)	283.4 (275.7)	0.223	285.9 (269.2)	331.4 (316.5)	0.009
Vegetables (g)	224.0 (181.6)	229.3 (197.2)	217.3 (172.8)	<0.001	222.4 (174.5)	227.9 (212.5)	0.104
Protein-rich foods (g)	209.3 (145.8)	184.6 (122.0)	242.7 (164.8)	<0.001	209.6 (145.6)	208.7 (150.1)	0.998
Ready-made meals (g)	94.76 (104.6)	74.0 (85.6)	120.1 (116.0)	<0.001	96.3 (109.5)	91.8 (91.8)	0.099
Dairy products (g)	177.7 (201.9)	179.7 (203.8)	174.5 (202.3)	0.210	175.5 (202.3)	181.8 (205.4)	0.677
Lipids (fats and oils) (g)	61.6 (51.6)	58.9 (48.4)	65.2 (58.1)	<0.001	61.6 (51.8)	61.1 (53.6)	0.088
Sugary products (g)	34.5 (41.2)	32.0 (37.6)	38.5 (43.5)	<0.001	33.6 (40.8)	35.8 (42.8)	0.282
Non-caloric beverages (mL)	1665 (999.3)	1668 (1105)	1660 (983.2)	0.455	1662 (992.1)	1673 (983.8)	0.875
Sugar-sweetened beverages (mL)	70.2 (236.7)	70.2 (233.3)	70.2 (236.6)	0.435	66.8 (236.6)	72.0 (233.3)	0.217
Alcoholic beverages (mL)	76.9 (157.4)	42.8 (93.8)	132.9 (226.1)	<0.001	85.4 (169.4)	59.5 (108.5)	0.038

*n* = number of participants included in the analysis, IQR = interquartile range, CS = constipation score (Q1 + Q2 + Q3), CS2 = Q4. ^ꝉ^ *t*-test was used for continuous variables. (All variables were log-transformed). * Benjamini–Hochberg correction was applied to all *p*-values; all *p*-values are displayed after this correction.

**Table 4 nutrients-14-00122-t004:** Distribution (median and IQR) of nutrient intake per day according to gender and constipation score groups.

Variables	Median (IQR)						
	ORISCAV-LUX 2 (*n* = 1431)	Women (*n* = 761)	Men (*n* = 670)	*p*-Value ^ꝉ,^*	CS1 (*n* = 1078)	CS2 (*n* = 353)	*p*-Value ^ꝉ,^*
Total energy (kcal)	2375 (1143)	2133 (920.5)	2687 (1210)	<0.001	2395 (1129)	2303 (1124)	0.291
Water (g)	3080 (1263)	3002 (1140)	3161 (1390)	0.004	3081 (1259)	3057 (1228)	0.692
Protein total (g)	89.0 (45.7)	79.7 (38.5)	102.3 (49.5)	<0.001	89.0 (45.3)	89.0 (47.4)	0.752
Vegetable protein (g)	26.9 (14.5)	24.7 (12.5)	29.4 (16.2)	<0.001	27.0 (14.6)	25.8 (14.4)	0.107
Animal protein (g)	60.1 (37.4)	53.2 (31.3)	69.4 (41.6)	<0.001	60.1 (37.3)	60.8 (37.7)	0.801
Total fats (g)	116.0 (65.5)	108.0 (56.1)	128.4 (68.0)	<0.001	117.4 (64.4)	113.7 (68.3)	0.423
SFA (g)	39.7 (23.3)	36.0 (20.1)	44.5 (25.3)	<0.001	40.1 (23.2)	38.3 (23.6)	0.499
MUFA (g)	46.9 (26.6)	43.0 (22.4)	52.7 (28.8)	<0.001	46.8 (27.0)	47.2 (26.8)	0.587
PUFA (g)	20.8 (13.6)	19.6 (12.5)	22.6 (14.6)	<0.001	20.9 (13.4)	20.2 (14.5)	0.269
LA (g)	17.3 (11.9)	16.2 (11.2)	18.8 (12.7)	<0.001	17.4 (11.7)	16.8 (12.6)	0.198
ALA (g)	1.80 (1.44)	1.68 (1.49)	1.93 (1.38)	<0.001	1.79 (1.47)	1.82 (1.43)	0.792
ARA (g)	0.19 (0.13)	0.17 (0.11)	0.22 (0.16)	<0.001	0.19 (0.13)	0.19 (0.14)	0.960
EPA (g)	0.20 (0.23)	0.18 (0.23)	0.21 (0.25)	<0.001	0.19 (0.23)	0.20 (0.24)	0.819
DPA (g)	0.08 (0.07)	0.07 (0.07)	0.08 (0.08)	<0.001	0.07 (0.07)	0.08 (0.07)	0.931
DHA (g)	0.28 (0.32)	0.26 (0.31)	0.29 (0.34)	<0.001	0.28 (0.31)	0.29 (0.32)	0.776
Cholesterol (mg)	356.2 (205.7)	320.7 (169.7)	395.4 (228.5)	<0.001	356.3 (206.4)	354.3 (199.9)	0.909
Total carbohydrates (g)	218.3 (110.4)	197.1 (92.4)	242.0 (119.2)	<0.001	219.7 (111.0)	213.8 (113.2)	0.339
Simple sugars (g)	101.3 (60.0)	96.6 (55.3)	109.2 (61.7)	<0.001	99.6 (58.4)	105.3 (68.5)	0.099
Added simple sugars (g)	28.7 (27.6)	26.1 (24.1)	32.4 (30.6)	<0.001	28.3 (26.9)	29.3 (28.4)	0.343
Starch (g)	103.7 (63.6)	90.6 (52.9)	119.1 (66.2)	<0.001	106.8 (63.6)	94.8 (60.0)	0.003
Total fiber (g)	23.2 (12.3)	22.9 (11.8)	24.0 (13.3)	0.008	23.3 (11.9)	23.1 (14.1)	0.975
Soluble fiber (g)	4.75 (2.48)	4.74 (2.46)	4.74 (2.49)	0.492	4.72 (2.38)	4.81 (2.81)	0.312
Insoluble fiber (g)	18.5 (10.1)	18.1 (9.5)	19.0 (10.7)	0.002	18.4 (9.6)	18.4 (11.5)	0.819
Alcohol (g)	5.6 (11.5)	3.3 (7.6)	9.3 (15.8)	<0.001	6.4 (12.3)	4.4 (8.7)	0.010
Beta-carotene (µg)	4978 (4125)	5157 (4472)	4713 (3843)	0.009	4960 (3956)	5062 (5132)	0.133
Vitamin A (retinol, µg)	476.8 (343.7)	425.1 (281.2)	547.7 (385.6)	<0.001	478.4 (336.0)	460.5 (382.4)	0.860
Vitamin D (µg)	5.1 (4.7)	4.8 (4.4)	5.6 (4.9)	<0.001	5.1 (4.6)	5.0 (4.8)	0.899
Vitamin E (α-tocopherol equivalents, mg)	18.3 (11.8)	16.7 (9.6)	20.3 (13.7)	<0.001	18.3 (11.5)	18.3 (12.5)	0.614
Vitamin C (mg)	145.6 (104.8)	150.9 (104.6)	142.4 (101.2)	0.159	143.4 (101.8)	157.5 (120.9)	0.009
Vitamin B1 (mg)	1.5 (0.8)	1.4 (0.6)	1.7 (0.9)	<0.001	1.5 (0.8)	1.5 (.08)	0.819
Vitamin B2 (mg)	1.8 (1.0)	1.6 (0.8)	2.1 (1.1)	<0.001	1.8 (0.9)	1.8 (1.0)	0.592
Vitamin B3 (mg)	23.0 (12.1)	20.1 (9.7)	26.2 (13.3)	<0.001	23.0 (11.8)	22.3 (12.3)	0.930
Vitamin B5 (mg)	5.8 (2.9)	5.3 (2.5)	6.5 (3.4)	<0.001	5.8 (2.8)	5.7 (3.2)	0.734
Vitamin B6 (mg)	2.4 (1.3)	2.2 (1.0)	2.6 (1.4)	<0.001	2.3 (1.2)	2.3 (1.3)	0.596
Vitamin B9 (µg)	350.8 (174.9)	340.3 (164.3)	363.7 (187.7)	<0.001	351.2 (170.7)	347.3 (190.4)	0.740
Vitamin B12 (µg)	6.3 (4.5)	5.5 (4.0)	7.4 (5.0)	<0.001	6.2 (4.4)	6.3 (4.9)	0.850
Calcium (mg)	930.5 (452.8)	893.8 (431.1)	984.1 (477.8)	<0.001	938.5 (450.3)	899.8 (474.8)	0.807
Iron (mg)	14.3 (6.8)	13.2 (6.0)	15.8 (7.8)	<0.001	14.3 (6.7)	14.2 (7.4)	0.622
Iodide (µg)	154.3 (80.3)	143.9 (72.2)	166.9 (86.8)	<0.001	155.2 (78.9)	152.1 (83.3)	0.846
Magnesium (mg)	372.1 (162.4)	352.5 (144.1)	405.6 (176.0)	<0.001	373.2 (162.8)	371.1 (167.3)	0.794
Sodium (mg)	3310 (1968)	2929 (1536)	3870 (2210)	<0.001	3339 (1978)	3229 (2005)	0.533
Potassium (mg)	3547 (1563)	3370 (1452)	3725 (1685)	<0.001	3544 (1505)	3562 (1749)	0.112
Phosphorous (mg)	1326 (622)	1205 (538)	1504 (712)	<0.001	1329 (624.6)	1295 (617.4)	0.671

*n* = number of participants included in the analysis, IQR = interquartile range, CS = constipation score (Q1 + Q2 + Q3), CS2 = Q4, (kcal) = kilocalories, (mg) = milligrams, (g) = grams, (µg) = micrograms, SFA = saturated fat, MUFA = monounsaturated fat, PUFA = polyunsaturated fat, LA = linolenic acid, ALA = alpha linoleic acid, ARA = arachidonic acid, EPA = eicosapentaenoic acid, DPA = docosapentaenoic acid, DHA = docosahexaenoic acid. ^ꝉ^ *t*-test was used for continuous variables (All variables were log-transformed). * Benjamini–Hochberg correction was applied to all *p*-values; all *p*-values are displayed after this correction.

**Table 5 nutrients-14-00122-t005:** Pearson coefficients of simple correlation between dietary factors and constipation score in age, gender, and BMI groups.

Variables	ORISCAV-LUX 2 (*n* = 1431)			
	All	Gender	Age Categories	BMI Categories
Total energy (kcal)	−0.058 *	–	–	–
Vegetable protein (g)	−0.059 *	–	–	–
Total fats (g)	−0.053 *	–	–	–
PUFA (g)	−0.054 *	–	−0.114 *^,e^	−0.129 *^,i^
Starch (g)	−0.092 **	–	−0.140 **^,e^	−0.093 *^,g^
Alcohol (g)	−0.083 **	–	−0.143 *^,a,d^ −0.108 *^,a,e^	−0.182 **^,h^
Vitamin C (mg)	0.054 *^,a^	–	–	–
Grains (g)	−0.078 **	–	−0.107 *^,f^	−0.084 *^,g^
Starchy vegetables (g)	−0.061 *	–	–	–
Fruits (g)	0.067 *^,a^	0.105 **^,a,b^	–	–
Vegetables (g)	0.062 *^,a^	0.076 *^,a,c^	–	–
Lipids (g)	−0.068 *	–	−0.127 *^,d^	−0.129 *^,i^
Alcoholic beverages (mL)	−0.083 **	–	−0.143 *^,a,d^	−0.121 *^,h^

*n* = number of participants included in the analysis, IQR = interquartile range, BMI = body mass index. (kcal) = kilocalories, (mg) = milligrams, (g) = grams, PUFA = polyunsaturated fatty acid. * = *p*-value 0.05, ** = *p*-value 0.01. ^a^ = no log10 transformation of data. ^b^ = men. ^c^ = women. ^d^ = Age group 2 (35–44.99). ^e^ = Age group 3 (45–54.99). ^f^ = Age group 4 (55–64.99). ^g^ = BMI group 1 (<24.99 kg/m^2^). ^h^ = BMI group 2 (25–29.99 kg/m^2^). ^i^ = BMI group 3 (>30 kg/m^2^).

**Table 6 nutrients-14-00122-t006:** Regression coefficients (beta, 95% CI) between food group intakes and constipation score based on linear regression modeling ^a^.

Variables	ORISCAV-LUX 2 (*n* = 1431)			
	Beta Non-Standardized	Beta Standardized	CI 95%	*p*-Value
Grains (g)	−0.616	−0.059	−1.183, −0.048	0.033
Vegetables (g)	0.545	0.044	-0.097, 1.186	0.096
Lipids (g)	−0.838	−0.063	−1.551, −0.126	0.021
Sugary products (g)	0.540	0.068	0.105, 0.974	0.015

*n* = number of participants included in the analysis. ^a^ Linear regression model adjusted for education level, job, testosterone serum level, sodium serum level; Total R was: 0.253.

**Table 7 nutrients-14-00122-t007:** Regression coefficients (beta, 95% CI) between nutrient intakes and constipation score based on the linear regression modelling ^a^.

Variables	ORISCAV-LUX 2 (*n* = 1431)			
	Beta Non-Standardized	Beta Standardized	CI 95%	*p*-Value
Total energy (kcal)	5.239	0.225	0.366, 10.112	0.035
Total fats (g)	−4.170	−0.207	−7.455, −0.885	0.013
Starch (g)	−2.912	−0.171	−4.466, −1.358	<0.001
Sodium intake (mg)	2.040	0.109	0.206, 3.874	0.029

*n* = number of participants included in the analysis. ^a^ Linear regression model adjusted for BMI, education level, job, testosterone serum level, sodium serum level; total R was: 0.258.

## Data Availability

The data presented in this study are available on request from the corresponding author. The data are not publicly available due to our institute rules and laws.

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
