# Peer review of "Association between Dietary Factors and Constipation in Adults Living in Luxembourg and Taking Part in the ORISCAV-LUX 2 Survey"

_nutrients, 2021, doi:10.3390/nu14010122_

Round 1

Reviewer 1 Report

(Reviewer 1) :

Generally, common causes of chronic constipation include a lack of fiber (inadequate consumption of fruits, vegetables, and other foods containing fiber), no sufficient drinking water or liquids. However, in your manuscript dietary fiber, fruit and vegetable intake did not show any association with constipation. Conflicting results were confirmed. This study encompasses an interesting topic, but it has some critical problems.

Major comments:

  1. Statistical analysis and data presentation are most important in the study of the relationship between dietary factors and constipation. Presenting all data in IQR reduces the readability of the paper. It would be better to present the collected data as the mean and standard deviation, and to display the median (SE) and IQR only when the data do not satisfy normality. In Table 2, it would be good to present statistical p-values such as demographic information and variables according to the characteristics of the subjects by sex.

    (For examples: Table 2, table 3, table 4)

  • Data are mean (SD) for continuous variables or
  • median ± SE, Interquartile range(Q1-Q3) if not normally distributed.

2. The purpose of this study was to investigate the relationship between dietary factor intake and constipation.

It is necessary to categorize the intake level for each food group and nutrients related to constipation into either tertile or quartile, and then perform multiple logistic regression after adjusting the related variables. In particular, the intake of grain, lipid rich foods, sugary products, sodium, total fats, starch, fiber, fruits, and vegetables, which are related to diet and nutrients related to the regression analysis results, was categorized. It is necessary to present the constipation days odds ratio (OR) for subjects (Q2, Q3) who consume a lot compared to the subjects (Q1) who consume the least by food group. Also, the P for trend of OR needs to be derived by applying the median for each food and nutrient intake level to the regression equation.

3. As a result of reviewing the questionnaire evaluation criteria for constipation scores (Reference 34), the diagnosis of global constipation symptoms corresponds to a case of 15 points or more in constipation patients. If the constipation score of the subject of this study was 5 or less, it was 82.8%. It is necessary to group the constipation symptom scores of the study subjects into two groups. For example, it can be effective to analyze the relationship between the intake of each food group and constipation by cutting the average constipation score of all subjects' constipation scores or dividing the median value into 2 groups and presenting the constipation odd ratio (OR). 95% confidence interval (CIs), and P trends values for constipation factors by intake of diets factors.

4. In this study, the backward elimination method was applied as the variable selection method in the multiple regression analysis, but it may not necessarily be the optimal model. As for the recent forward selection methods to compensate for these shortcomings, it seems that it can be a reliable model to analyze by applying stepwise regression methods, which is a method combined with the backward elimination method.

5. In general, the prevalence of constipation is higher in women than in men, so it is necessary to consider a stratified analysis method by gender.

6. Presenting all the data in this study as Median and IQR without providing statistics for general, food group, and nutrient intake is not readable for understanding constipation risk factors.

(For examples):  Statistical difference in diet and nutrient intake according to 2 groups of constipation score

Minor

  1. Table 1 has already been presented in the preceding reference (34), so it needs to be deleted. This is unnecessary. The title for table 1 seems strange (p 8, line 271) Table 1-> Table 2 should be modified.

2. Rather than presenting all nutrients in Table 4, it is necessary to select variables related to constipation. (example : Fatty acid related details)

3. It would be better to move Figure 3 to a supplementary materials.

Author Response

Generally, common causes of chronic constipation include a lack of fiber (inadequate consumption of fruits, vegetables, and other foods containing fiber), no sufficient drinking water, or liquids. However, in your manuscript dietary fiber, fruit, and vegetable intake did not show any association with constipation. Conflicting results were confirmed. This study encompasses an interesting topic, but it has some critical problems.

Major comments:

  1. Statistical analysis and data presentation are most important in the study of the relationship between dietary factors and constipation. Presenting all data in IQR reduces the readability of the paper. It would be better to present the collected data as the mean and standard deviation and to display the median (SE) and IQR only when the data do not satisfy normality. In Table 2, it would be good to present statistical p-values such as demographic information and variables according to the characteristics of the subjects by sex. (For example Table 2, Table 3, and table 4). Data are mean (SD) for continuous variables or median ± SE, Interquartile range (Q1-Q3) if not normally distributed.

Reply: We agree with the reviewer that the best way to present continuous variables is reporting mean±SD, however as the data did not satisfy the criteria of normality, we reported median along with IQRs, as proposed. In addition, also as proposed by the reviewer, we have revised Tables 2, 3, and 4 and provided p-values according to the gender and constipation score categories (based on quartiles).

  1. The purpose of this study was to investigate the relationship between dietary factor intake and constipation. It is necessary to categorize the intake level for each food group and nutrients related to constipation into either tertiles or quartile, and then perform multiple logistic regression after adjusting the related variables. In particular, the intake of grain, lipid-rich foods, sugary products, sodium, total fats, starch, fiber, fruits, and vegetables, which are related to diet and nutrients related to the regression analysis results, was categorized. It is necessary to present the constipation days odds ratio (OR) for subjects (Q2, Q3) who consume a lot compared to the subjects (Q1) who consume the least by food group. Also, the P for a trend of OR needs to be derived by applying the median for each food and nutrient intake level to the regression equation.

Reply: We appreciate the reviewer’s comment. We have now revised Tables 3 and 4 and reported macro- and micro-nutrients and food group intakes based on the quartile grouping of constipation scores. Since almost 80% of the participants in our study reported no problems (their constipation score was zero), in practice, it was not feasible to classify them according to tertiles or quartiles. This is because, for example, in the first three quartiles, all people would have a score of zero. Therefore, the best solution seemed to be to divide them into quartiles and merge three of them (Q1, 2, and 3) and compare them with Q4 (as now presented).

  1. As a result of reviewing the questionnaire evaluation criteria for constipation scores (Reference 34), the diagnosis of global constipation symptoms corresponds to a case of 15 points or more in constipation patients. If the constipation score of the subject of this study was 5 or less, it was 82.8%. It is necessary to group the constipation symptom scores of the study subjects into two groups. For example, it can be effective to analyze the relationship between the intake of each food group and constipation by cutting the average constipation score of all subjects' constipation scores or dividing the median value into 2 groups and presenting constipation as the odd ratio (OR). 95% confidence interval (CIs) and P trends values for constipation factors by intake of diets factors.

Reply: We agree with the reviewer. We have now divided individuals into two groups (quantiles) based on constipation scores and performed logistic regression analyzes. The results are presented in lines 323 ff..

  1. In this study, the backward elimination method was applied as the variable selection method in the multiple regression analysis, but it may not necessarily be the optimal model. As for the recent forward selection methods to compensate for these shortcomings, it seems that it can be a reliable model to analyze by applying stepwise regression methods, which is a method combined with the backward elimination method.

Reply: We agree with the reviewer. We have performed several of these analyzes in order to be more confident and to make sure that any possible association is found.The resultant models based on the backward elimination and combined with additional forward selection resulted in retaining the most significant number of parameters in the model.

  1. In general, the prevalence of constipation is higher in women than in men, so it is necessary to consider a stratified analysis method by gender.

Reply: We thank the reviewer for the comment. We have redone all tests also stratified for gender.

  1. Presenting all the data in this study as Median and IQR without providing statistics for general, food group, and nutrient intake is not readable for understanding constipation risk factors. (For examples):  Statistical difference in diet and nutrient intake according to 2 groups of constipation score.
  2. Reply: We thank the reviewer for the comment. We have now analyzed all tests based on 2 groups of constipation scores (see tables 3 and 4).

Minor comments:

  1. Table 1 has already been presented in the preceding reference (34), so it needs to be deleted. This is unnecessary. The title for table 1 seems strange (p 8, line 271) Table 1-> Table 2 should be modified.

Reply: We agree with the reviewer. We have revised the title of the table. However, we preferred to keep the table presenting the scoring system, as we perceived it important (and assuredly more convenient for the reader) to have a better interpretation of the scoring algorithm.

  1. Rather than presenting all nutrients in Table 4, it is necessary to select variables related to constipation. (Example: Fatty acid-related details).

Reply: We thank the reviewer for the comment. We think the information provided can be useful for better interpretation of the results by readers, especially in sight of the rather explorative approach, i.e. not upfront excluding nutrients.

  1. It would be better to move Figure 3 to supplementary materials.

Reply: We thank the reviewer for the comment. Since the difference between constipation scores in men and women as well as age groups is notable, we strongly feel that this figure gives the reader an overview of gender and age differences and it is better to be in the original text rather than in supplementary materials.

References:

  1. MCCREA GL, MIASKOWSKI C, STOTTS NA, MACERA L, HART SA, VARMA MG. Review article: self-report measures to evaluate constipation. Alimentary Pharmacology & Therapeutics. 2008;27(8):638-48.
  2. Agachan F, Chen T, Pfeifer J, Reissman P, Wexner SD. A constipation scoring system to simplify evaluation and management of constipated patients. Diseases of the colon & rectum. 1996;39(6):681-5.

Reviewer 2 Report

If people over 81 were enrolled, why were people over 79 excluded?

Please specify what it means “Not recruited”?

Why were 13 cases excluded?

The question “Abdominal pain” refers to during evacuation or generally before evacuation? Has this clearly been presented to the participants.

Why is the score always 0-4? It indicates that “Frequency of bowl movement (less than once a month) =4 has the same importance as “Minutes spend in lavatory >30 minutes =4” even though clearly the two not have the same importance. This should be more carefully thought about and discussed. Also use of laxative (2 points) is similar to 20 minutes on the toilet. Where it should be clear that use of laxatives has a far higher weighting. Much has to do with the constipation score. By weighting some factors over others – the authors can highlight effects. Here, simply equating all effects equally means that some smaller effects are overly represented. A clear discussion in the creation of the scoring sheet and justification on the scoring (for each section) is therefore required.

Isn’t “job” similar to “physical exercise”? 

Author Response

  1. If people over 81 were enrolled, why were people over 79 excluded?

Reply: We thank the reviewer for the comment. In fact, the study protocol stated that people could enter the study until the age of 79. However, since this study was a continuation of the first wave (taking place 10 years earlier), a few participants in the second wave were over 79 years old. We have clarified this now in the current version of the manuscript, please see lines 95-98.

  1. Please specify what it means “Not recruited”? Why were 13 cases excluded?

Reply: We have now revised the figure footnote to be clearer. In fact, these cases were excluded because of reasons for moving abroad, physical disability, or language incapacity (old age).

  1. The question “Abdominal pain” refers to during evacuation or generally before evacuation? Has this clearly been presented to the participants?

Reply: Based on the original literature this refers to general “abdominal pain” and in our study educated and trained nurses asked these questions to the participants.

  1. Why is the score always 0-4? It indicates that “Frequency of bowl movement (less than once a month) =4 has the same importance as “Minutes spend in lavatory >30 minutes =4” even though clearly the two not have the same importance. This should be more carefully thought about and discussed. Also use of laxative (2 points) is similar to 20 minutes on the toilet. Where it should be clear that use of laxatives has a far higher weighting. Much has to do with the constipation score. By weighting some factors over others – the authors can highlight effects. Here, simply equating all effects equally means that some smaller effects are overly represented. A clear discussion in the creation of the scoring sheet and justification on the scoring (for each section) is therefore required.

Reply: We agree with the reviewer that the choice of weighing the various questions may appear somewhat subjective. However, this questionnaire is based on a published questionnaire, which has been validated in a trial with constipated patients (Western country) against objective physiological findings. Furthermore, the publication and the questionnaire have been employed in many other studies (it has been cited 269 times).

Although we agree with the reviewer that one may have weighted the questions differently, as the scoring algorithm has been used in other studies (without weighting, e.g. (1)(2)) and its relationship/correlation to biological outcomes were investigated in the original study, changing the weighting may damage its original validity. Perhaps studies with other designs (validity/reliability/calibration) in the future could further address the question of whether weighting may help the accuracy of the score, but this was beyond the scope of the present study. We have mentioned this aspect in the study limitations’ section; please see lines 461 ff.

  1. Isn’t “job” similar to “physical exercise”? 

Reply: We agree with the reviewer that there are similarities between them but they are not interchangeable. Physical activity referred also to additional exercise, also done outside the job.  Thus, it was deemed superior to separate and analyze them individually.

Reviewer 3 Report

In this article, data from the 2016-2017 ORISCAV-LUX 2 study was analyzed. The study used a questionnaire-based constipation score and further assessed confounders such as physical activity (via accelerometers) and serum/urine indicators, including C-reactive protein, thyroid-stimulating hormone, free triiodothyronine, and free thyroxine hormones. A total of 1431 participants were included in the study. The authors found, in a food-group based regression model, a negative association between higher constipation score and grains and lipid-rich foods. Further, a positive association was found for a higher constipation score and sugary products. However, no association was found between constipation and fruits and vegetables or dietary fiber. While the results of this study are not unexpected, they remain important to publish.

Introduction: Brief and covered the research background very well.

Materials and Methods: Nicely clarified and no issues with the statistical analysis found.

Results: I would remove the hash marks in Table 2, and edit the variables for better clarity.

Discussion: Appropriate for the results.

Author Response

In this article, data from the 2016-2017 ORISCAV-LUX 2 study was analyzed. The study used a questionnaire-based constipation score and further assessed confounders such as physical activity (via accelerometers) and serum/urine indicators, including C-reactive protein, thyroid-stimulating hormone, free triiodothyronine, and free thyroxine hormones. A total of 1431 participants were included in the study. The authors found, in a food-group based regression model, a negative association between higher constipation score and grains and lipid-rich foods. Further, a positive association was found for a higher constipation score and sugary products. However, no association was found between constipation and fruits and vegetables or dietary fiber. While the results of this study are not unexpected, they remain important to publish.

Introduction: Brief and covered the research background very well.

Materials and Methods: Nicely clarified and no issues with the statistical analysis found.

Results: I would remove the hash marks in Table 2, and edit the variables for better clarity.

Discussion: Appropriate for the results.

Reply: We thank the reviewer for the encouraging comments; these were much appreciated.

Round 2

Reviewer 2 Report

May be accepted in its current form.